# Small variant benchmark from a complete assembly of X and Y chromosomes

Justin Wagner[1,18], Nathan D. Olson [1,18], Jennifer McDaniel [1], Lindsay Harris[1], Brendan J. Pinto[2], David Jáspez [3], Adrián Muñoz-Barrera [3], Luis A. Rubio-Rodríguez[3], José M. Lorenzo-Salazar [3], Carlos Flores [3,4,5,6], Sayed Mohammad Ebrahim Sahraeian[7], Giuseppe Narzisi [8], Marta Byrska-Bishop [8], Uday S. Evani[8], Chunlin Xiao[9], Juniper A. Lake [10], Peter Fontana[11], Craig Greenberg[11], Donald Freed [12], Mohammed Faizal Eeman Mootor [13], Paul C. Boutros [13], Lisa Murray[14], Kishwar Shafin [15], Andrew Carroll [15], Fritz J. Sedlazeck [16], Melissa Wilson[17] & Justin M. Zook [1] ✉

The sex chromosomes contain complex, important genes impacting medical phenotypes, but differ from the autosomes in their ploidy and large repetitive regions. To enable technology developers along with research and clinical laboratories to evaluate variant detection on male sex chromosomes X and Y, we create a small variant benchmark set with 111,725 variants for the Genome in a Bottle HG002 reference material. We develop an active evaluation approach to demonstrate the benchmark set reliably identifies errors in challenging genomic regions and across short and long read callsets. We show how complete assemblies can expand benchmarks to difficult regions, but highlight remaining challenges benchmarking variants in long homopolymers and tandem repeats, complex gene conversions, copy number variable gene arrays, and human satellites.

The complete human karyotype includes 22 pairs of autosomes and two sex chromosomes (X and Y). The unique biology of the X and Y chromosomes makes their analysis more difficult than the autosomes in some ways. Indeed, the X and Y chromosomes contain many medically relevant genes, as well as very challenging repetitive regions[1–5]. Chromosomes X and Y mostly have distinct sequences, but two pseudoautosomal regions (PARs) experience crossover events similar to autosomes, and the recently X-transposed region (XTR) retains relatively high sequence identity between X and Y[6]. Benchmark sets from well-characterized samples are important for understanding

[1]Material Measurement Laboratory, National Institute of Standards and Technology, 100 Bureau Dr., Gaithersburg, MD, USA. [2]Center for Evolution & Medicine and School of Life Sciences, Arizona State University, Tempe, AZ 85281 USA - Department of Zoology, Milwaukee Public Museum, Milwaukee, WI, USA. [3]Genomics Division, Instituto Tecnológico y de Energías Renovables (ITER), Granadilla de Abona, Spain. [4]CIBER de Enfermedades Respiratorias (CIBERES), Instituto de Salud Carlos III, Madrid, Spain. [5]Research Unit, Hospital Universitario Nuestra Señora de Candelaria, Instituto de Investigación Sanitaria de Canarias, Santa Cruz de Tenerife, Spain. [6]Facultad de Ciencias de la Salud, Universidad Fernando de Pessoa Canarias, Las Palmas de Gran Canaria, Spain. [7]Roche Sequencing Solutions, Santa Clara, CA, USA. [8]New York Genome Center, New York, NY, USA. [9]National Center for Biotechnology Information, National Library of Medicine, National Institutes of Health, Bethesda, MD, USA. [10]Pacific Biosciences, Menlo Park, CA, USA. [11]Information Technology Laboratory, National Institute of Standards and Technology, 100 Bureau Dr. Mailstop 8940, Gaithersburg, MD, USA. [12]Sentieon Inc., San Jose, CA, USA. [13]Department of Human Genetics, University of California Los Angeles, Los Angeles, CA, USA. [14]Illumina, Cambridge, United Kingdom. [15]Google Inc, 1600 Amphitheatre Pkwy, Mountain View, CA, USA. [16]Baylor College of Medicine Human Genome Sequencing Center, Houston, TX, USA. [17]Center for Evolution & Medicine and School of Life Sciences, Arizona State University, Tempe, AZ, USA. [18]These authors contributed equally: Justin Wagner, Nathan D. Olson. ✉e-mail: justin.zook@nist.gov

variant call accuracy. Previous Genome in a Bottle Consortium (GIAB) benchmarks excluded the X and Y chromosomes due to their mostly-hemizygous (i.e., haploid) nature in half the population[7,8], which requires customized variant calling methods[9]. However, recently, GIAB has developed approaches to form variant benchmarks by aligning long-read assemblies to the refs. [10,11], enabling the generation of benchmarks for more challenging regions and variants. Here, we create benchmarks that include challenging variants and regions using complete, polished de novo assemblies of the X and Y chromosomes in the GIAB Personal Genome Project[12] Ashkenazi Jewish son HG002 from the Telomere-to-Telomere (T2T) Consortium[1,13]. We also pilot a systematic approach to evaluate benchmarks to exclude problematic regions and ensure the final benchmark reliably identifies errors in a variety of genome contexts.

## Results

### Benchmark generation

We created a small variant benchmark for regions that could be confidently assembled, unambiguously aligned to GRCh38, and where small variant benchmarking (such as hap.py, vcfeval, and vcfdist) are reliably able to compare variants with different representations (Fig. 1). Based on previous GIAB work[8], we included only loci with exactly one contig aligning from each haplotype in the X PAR or where one contig

aligns from the X or Y assembled haplotype in the X and Y non-PAR, respectively (this is the dip.bed file output from dipcall). Beyond excluding regions outside the dipcall bed file, we found that it was important to exclude additional regions that may contain complex structural variation where the assembly cannot be unambiguously aligned to the reference, as well as assembly errors identified while developing the benchmark set (Table 1 and Fig. 2). Specifically, we excluded structural variants at least 50 bp in size and associated repeats, large repeats (segmental duplications, tandem repeats longer than 10 kbp, and satellites) that have any breaks in the assembly-assembly alignment, and regions around gaps in GRCh38[14]. Based on the evaluation described in the next section, we also excluded some homopolymers and tandem repeats with potential assembly errors, as well as sites affected by a bug in dipcall that misses some deletions adjacent to insertions around complex variants.

### Evaluation of the draft benchmark

We evaluated the draft benchmark to test GIAB's criteria for Reference Materials to be fit-for-purpose, which is the reliable identification of errors in diverse variant callsets[15]. In this work, we piloted a method for evaluating machine learning systems called Active Evaluation, which takes advantage of stratifying a dataset and estimates a confidence interval for a system's

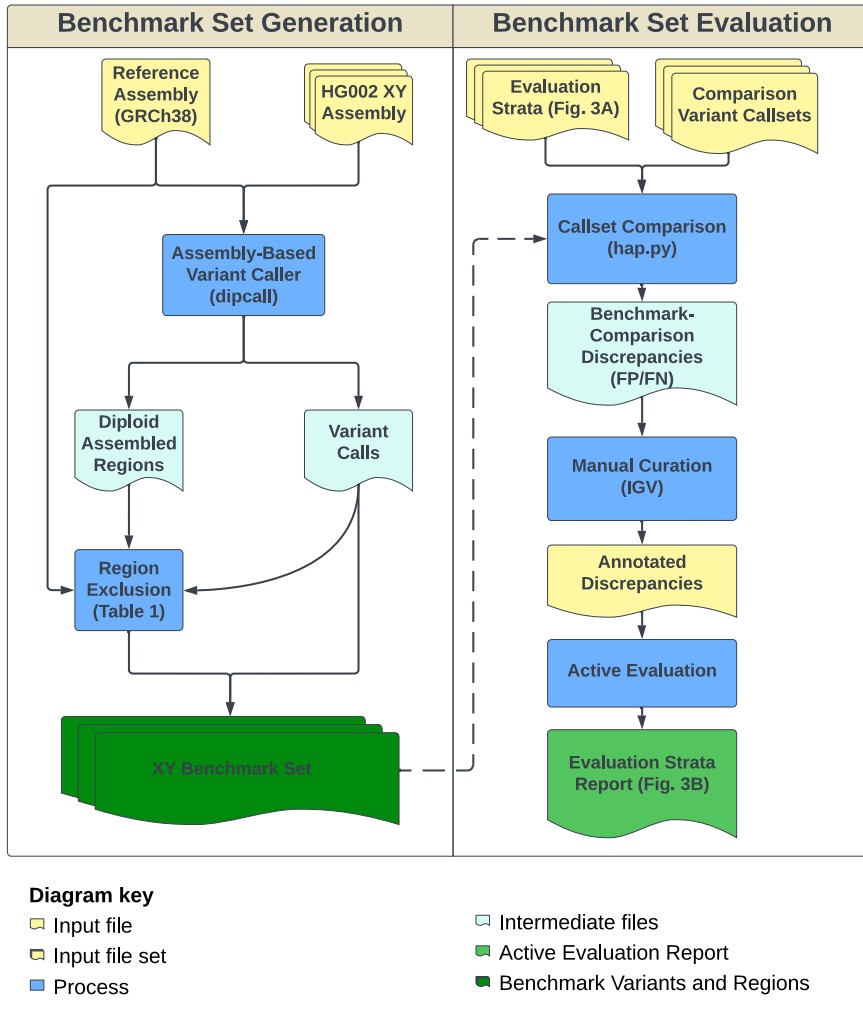

**Fig. 1 | Benchmark set generation and evaluation process.** To generate the benchmark, a polished complete assembly of the HG002 X and Y chromosomes was aligned to GRCh38 to create benchmark variant calls. The benchmark regions only include regions with the expected one-to-one alignment between each haplotype and the reference, and additional regions were excluded that are problematic for variant representation or small variant comparison (Table 1). An active evaluation process was used to demonstrate that the benchmark reliably identifies errors across a variety of comparison variant callsets from different technologies.

**Table 1 | Regions excluded from the assembly-based benchmark**

| Exclusion | Excluded bp | Resulting bp included |
|---|---|---|
| Dipcall regions | 40,069,198 | 173,199,112 |
| Homopolymers discordant with Element | 76,272 | 173,122,840 |
| Manually identified dipcall bugs at adjacent indels | 299 | 173,122,541 |
| Discrepancies with HPRC hifiasm HG002 assembly in tandem repeats | 92,490 | 173,030,051 |
| Automatically identified dipcall bugs using HiFi-DV calls | 1527 | 173,028,524 |
| Perfect or imperfect homopolymers longer than 30 bp in GRCh38 | 2,360,813 | 172,888,123 |
| Segmental duplications with break in dipcall bed | 7,123,154 | 168,563,138 |
| Tandem repeats longer than 10 kb with break in dipcall bed | 3,982,086 | 166,950,649 |
| Satellite repeats with break in dipcall bed | 3,970,501 | 166,770,555 |
| 15 kb around gaps in GRCh38 | 55,560,636 | 166,005,981 |
| GRCh38 self chain alignments with break in dipcall bed | 7,263,238 | 164,949,994 |
| 15 kb flanking regions around break in dipcall bed | 2,005,881 | 164,920,498 |
| SVs ≥50 bp and any overlapping tandem repeats and homopolymers | 7,377,984 | 163,075,893 |
| Errors and questionable variants and regions identified during curation | 1,526,347 | 161,549,546 |

GRCh38 X and Y chromosome lengths are 213,268,310 bp with gaps and 181,308,082 without gaps.
These regions contain potential errors in the assembly, large structural variation, ambiguous alignments of the assembly to the reference, or are incompatible with current small variant benchmarking tools.

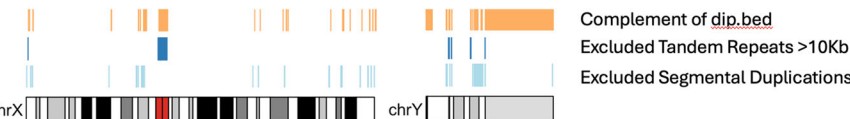

**Fig. 2 | Large regions excluded from the benchmark.** We exclude regions if they do not have the expected 1:1 alignment of the assembly to the reference, such as large gaps in GRCh38 and regions with very large structural variation in HG002 relative to GRCh38 (Complement of dip.bed). We also exclude large tandem repeats and segmental duplications if they contain a break in the assembly to reference alignment. Table 1 contains the full list of regions excluded from the benchmark.

performance. We defined 15 stratifications based on homopolymer length, difficult-to-map regions, and pseudoautosomal regions, as well as SNVs vs indels and putative false positives (FP) vs false negatives (FN) (Fig. 3).

We performed manual curation of a subset of putative FP and FN calls resulting from a comparison of a variety of callsets from different technologies against the draft benchmark (Supplementary Data 1). This effort was similar to the evaluation process of the v4.2.1[7] and Challenging Medically Relevant Gene (CMRG)[8] GIAB benchmarks. We updated this approach using a more focused sampling of different subsets, or evaluation strata, of FPs and FNs. We then used an Active Evaluation approach (https://github.com/usnistgov/active-evaluation) to estimate a score indicating how often the benchmark was determined to be correct by manual curation, as well as confidence intervals for each score. In using this approach, the overall goal is that the benchmark has a score of greater than 0.5 with 95% confidence (i.e., based on manual curation, the benchmark was estimated to be correct for more than 50% of the FPs and FNs with 95% confidence. Overall, as seen in Supplementary Data 2 across all callsets the lower 95% confidence interval is above 0.5 and in many of the callsets is significantly higher. As the lowest value is 0.55, all system 95% lower confidence intervals are above 0.50, meaning that if we were to curate the entire population it is likely that the benchmark is determined to be correct more than 50% of the time. This provides users evidence and well-defined confidence intervals that the benchmark is fit for purpose to reliably identify errors in each callset.

We found that, overall, the draft benchmark reliably identifies errors across stratifications and callsets (Figs. 3 and Supplementary Data 1 and 2). One notable exception was for indel FPs and FNs in an Element callset, where most of the putative FPs and FNs were errors in the benchmark in long homopolymers. Element's avidity sequencing

has been demonstrated to have high accuracy in homopolymers[16]. Therefore, during our evaluation effort, we refined the draft benchmark to create v1.0 by excluding additional region types, such as certain homopolymers and tandem repeats in Table 1. We also excluded all regions containing variants identified as incorrect or 'unsure' during manual curation.

In addition to manual curation, we performed long-range PCR followed by Sanger sequencing on a subset of challenging genes overlapping segmental duplications in chromosomes X and Y as a means of orthogonal validation of the benchmark variants. We confirmed a total of 181 variants in segmental duplications in 10 genes: *ARHGAP6, CLIC2, CSAG1, F8, IKBKG, NXF5, OPN1LW, OPN1MW, SAGE1, SLC6A8, SLC6A14, TMLHE*. Only one variant out of the 181 variants appeared to be contradicted by Sanger sequencing. However, in this case the assembly was clearly supported by long reads and the reason for the different Sanger result was unclear. The variant confirmation results and PCR conditions are detailed in Supplementary Data 3.

### The benchmark includes many challenging regions

The resulting benchmark contains substantially more challenging variants and regions than previous benchmarks. It includes 94% of chromosome X and 63% of chromosome Y, after excluding the 0.7% of chromosome X and 53% of chromosome Y missing from GRCh38. Of 294 medically relevant genes on these chromosomes[8], 270 are >90% included; of which 251 are >99% included in the small variant benchmark regions. 87 included genes are frequently tested by clinical laboratories[8] and the only frequently tested genes included <90% are *OPN1LW, OPN1MW, IKBKG*, and *G6PD*, all of which are excluded from our small variant benchmark due to large structural variation but are resolved by the assembly[17]. The benchmark includes some challenging regions like 68% of segmental duplications, and 99% of the XTR.

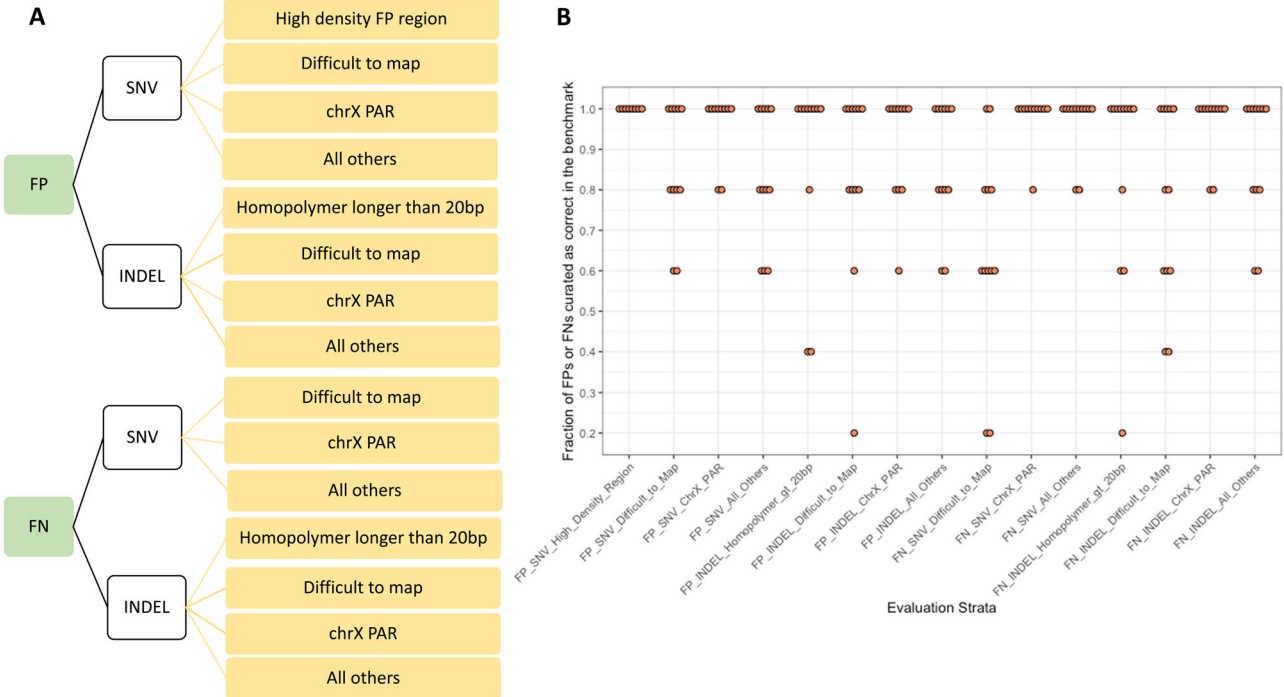

**Fig. 3 | Active evaluation process to demonstrate the benchmark reliably identifies errors across stratifications. A** Strata descriptions used during evaluation. Putative False Positive and False Negative SNVs and INDELs were categorized into different genomic regions. These include the High Density FP region shown in Supplementary Fig. 1, difficult to map regions, the PAR of chrX, homopolymers longer than 20 bp, and all others outside these regions. **B** Evaluation of

benchmark against a variety of sequencing technologies and variant calling methods. Each point represents curation results for each variant callset against a draft version of the benchmark set with the points below 0.5 as support for excluding regions from the v1.0 benchmark. Additionally, all variants curated as unsure or incorrect in the benchmark were excluded from the v1.0 benchmark regions.

**Table 2 | F1 scores for 36x Sequel II HiFi 14 kb reads with DeepVariant from the 2020 precisionFDA challenge and 48x Revio HiFi 17 kb reads with DeepVariant from 2023 against three recent GIAB benchmarks: the mapping-based v4.2.1 small variant benchmark for autosomes, the small variant benchmark for 273 challenging medically-relevant autosomal genes (CMRG) in autosomes, and the XY benchmark in this work**

| | | Long read (2020) | | | Long read (2023) | | |
|---|---|---|---|---|---|---|---|
| Variant Type | Region | v4.2.1 | CMRG | XY | v4.2.1 | CMRG | XY |
| SNV | All benchmark | 1.000 | 0.981 | *0.935* | 1.000 | 0.985 | *0.981* |
| SNV | Segmental duplications | 0.990 | 0.927 | *0.788* | 0.993 | 0.937 | *0.858* |
| SNV | Coding regions | 0.999 | 0.995 | *0.892* | 0.999 | 0.999 | *0.925* |
| INDEL | All benchmark | 0.996 | 0.967 | *0.741* | 0.982 | 0.932 | *0.856* |
| INDEL | TRs | 0.997 | 0.934 | *0.651* | 0.997 | 0.932 | *0.898* |
| INDEL | Homopolymers longer than 6 bp | 0.994 | 0.964 | *0.716* | 0.960 | 0.883 | *0.777* |
| INDEL | Not in Homopolymers longer than 6 bp | 0.998 | 0.969 | *0.764* | 0.998 | 0.969 | *0.935* |
| Insertions > 15 bp | All benchmark | 0.997 | 0.919 | *0.510* | 0.996 | 0.907 | *0.750* |

These do not reflect current accuracy, but exemplify how performance metrics can depend substantially on the difficulty of variants and regions included in the benchmark.

Although this benchmark excludes very challenging regions, it correctly identifies many errors in challenging regions, including some not well-represented in previous benchmarks. Table 2 describes how the benchmark helps assess performance for challenging variants and regions to understand differences between an older long-read HiFi variant callset from the 2020 precisionFDA Challenge, which was state-of-the-art at the time, and a more recent long-read HiFi variant callset from 2023[18,19]. For previous HG002 benchmarks sets, we use the assembly-based autosomal CMRG benchmark[8], which includes 273 challenging medically relevant genes, and the older GIAB v4.2.1 small variant benchmark[7], which used mapping-based approaches except in the major histocompatibility complex region. Both variant callsets

perform substantially worse against the XY benchmark relative to CMRG and especially v4.2.1. This reduction in performance is particularly evident for SNVs in segmental duplications, indels longer than 15 bp, and indels in tandem repeats and homopolymers. More challenging segmental duplications are included in the X and Y chromosomes relative to previous benchmarks, including those with gene conversions like *MAGEA3*, *CSAG2*, and *MAGEA6* described below. The XY benchmark also contains more challenging coding regions of genes, even compared to the previous challenging medically relevant genes benchmark, since both callsets have more than 10 times higher error rates for XY vs CMRG. The lower F1 score for insertions > 15 bp seen against the XY benchmark can be attributed to a higher rate of

**Fig. 4 | Visualization of gene conversion events included in the benchmark.** Visualization of genes in a region containing an inverted segmental duplication pair using pangene, a pangenome visualization tool[21].

genotype errors for longer indels, which are enriched in tandem repeats. By including more challenging tandem repeats in the XY benchmark, we show that indels in TRs are more accurate in the more recent callset, whereas accuracies for older and more recent callsets are similar when using the v4.2.1 or CMRG benchmarks. Interestingly, there are more homopolymer errors for v4.2.1 and CMRG in the more recent callset, probably because the more recent data from the Revio has longer reads and fewer passes around each molecule. Full stratified benchmarking results for many types of difficult genomic regions for each benchmark are in Supplementary Data 4–9. The benchmark regions also have four times lower variant density on Y (170/Mbp) compared to X (746/Mbp) and the autosomes (1530/Mbp on both haplotypes), as expected from previous work[20]. Lower variant density can result in lower precision because it decreases the number of true variants in the denominator.

By using alignments of the complete assembly to the GRCh38 reference, we included some gene conversion-like events in the benchmark. In these regions of HG002, the sequence of one GRCh38 region is replaced by the sequence of another similar GRCh38 region. For example, pangene, a pangenome visualization tool[21], shows that the genes *MAGEA6* and *MAGEA3* are swapped and inverted in HG002 and most Human Pangenome Reference Consortium (HPRC) samples relative to GRCh38[22] (Figs. 4 and Supplementary Fig. 2). The gene *CSAG2* is in the same segmental duplication in GRCh38, and in HG002 the region more closely matches *CSAG3*. Unlike *MAGEA3* and *MAGEA6*, which are swapped, HG002 also still has *CSAG3* in its location on GRCh38, so it is more similar to a stereotypical intrachromosomal gene conversion event. Even further towards the middle of these segmental duplications, *MAGEA2* and *MAGEA2B* are identical in GRCh38, and are also almost identical to each other in HG002, but their sequences in HG002 are diverged from GRCh38. In the middle of this pair of segmental duplications, *CSAG1* and *MAGEA12* have some SNVs but do not appear inverted or converted.

Beyond gene conversion-like events, the benchmark includes other challenging variants and regions. For example, Supplementary Fig. 3 shows an example of complex variants in a tandem repeat in the PAR, where the assembly resolves multiple phased variants on each haplotype within the repeat, which can be challenging for mapping-based approaches. Another challenging region for variant calling in the benchmark is a 856 kbp region on the Y chromosome with a high density of FPs due to extra divergent copies of the sequence in HG002 but not GRCh38, shown in Supplementary Fig. 1. The assembly resolves the correct sequence in this region, whereas mapping-based methods contain many FPs due to mismapped reads.

**Remaining challenging regions excluded from the benchmark**
We excluded other challenging regions due to errors in the HG002 assembly and ambiguous assembly-assembly alignments around large, complex SVs. We highlight two examples of challenging segmental duplications on chromosome Y that we excluded from the benchmark. The first region is a small known inversion error in the HG002 assembly[1] flanked by a pair of very large segmental duplications (chrY:17,455,804-17,951,260 and chrY:17,954,718-18,450,201). The intervening sequence between these regions is incorrectly inverted and the flanking segmental duplications also contain some small errors. Thus, we excluded the inversion and flanking segmental duplications (chrY:17,455,804-18,450,201).

The second set of regions excluded involves two different assembly-assembly alignment challenges posed by the Testis Specific Protein Y-linked (*TSPY*) gene family. The *TSPY* gene family is highly polymorphic in copy number[23] and has been implicated in risk of infertility and cancer[24,25]. The first challenge is that segmental duplications including the *TSPY2* gene were found to be swapped with their homologous sequences about 4 Mbp downstream in HG002 and most individuals relative to GRCh38[23]. We found that standard assembly-based variant calling methods such as dipcall, used in our work, tend to align the assembly contiguously rather than breaking the alignment and aligning *TSPY2* in HG002 to *TSPY2* in GRCh38. This results in variant calls where *TSPY2* is mostly deleted at its location in GRCh38 and inserted in its location in HG002. It is not standardized whether the variants should be called in this way or whether *TSPY2* in HG002 should be aligned to *TSPY2* in GRCh38, resulting in large translocations and smaller variants. Therefore, we chose to exclude these regions from the current benchmark bed file. Since there are multiple segmental duplication pairs annotated in this region of GRCh38 that appear to be swapped in HG002, we excluded the segmental duplication pairs in these regions: chrY:6,234,812-6,532,742 and chrY:9,628,425-9,919,592. In addition to *TSPY2* moving about 4 Mbp, the nearby gene *TTTY22*, as well as psuedogenes *RBMY2NP*, *RMBY2GP* and some other *TTTY* paralogs also swap positions between the segmental duplications chrY:6,234,812-6,532,742 and chrY:9,628,425-9,919,592. Interestingly, all of these genes are annotated differently on T2T-Y by CAT+Liftoff and RefSeq. Where RefSeq seems to match gene sequences, CAT+Liftoff tries to match positions even if the gene sequences differ.

In addition to the *TSPY2* challenge, the *TSPY* gene array is expanded substantially in HG002 relative to GRCh38, with 46 copies in HG002 versus 9 copies in GRCh38, which has a 40 kbp gap (see "Ampliconic genes in composite repeats" in Ref 1). In this case, no standards exist for whether this should be represented as one or more very large insertions alongside smaller variants, as a copy number variant, as a tandem repeat expansion, or some alternate representation. For these reasons, we excluded the *TSPY* gene array from the benchmark.

## Discussion
These results demonstrate the ongoing need for better benchmarking tools for tandem repeats, large duplications, and complex SVs. Even where assemblies are correct, there is a need for improved assembly-assembly alignment methods along with standards for representing and comparing variants in regions that do not have 1:1 alignment between assembly and ref. 26. The clearest example of this is *TSPY2*, excluded in this current benchmark. Also, a small number of structural errors remain in these complete X and Y assemblies, such as the inversion on chromosome Y that is polymorphic in the population as identified by Ref 1. Additionally, there are still many errors in homopolymers and dinucleotide tandem repeats in current assemblies which are not completely captured by the kmer-based methods typically used to assess assembly base-level accuracy. Another consideration is mosaic variants, or low frequency variants present in a subpopulation of cells, which we are in the process of identifying and characterizing. We acknowledge an additional limitation of the HG002 XY v1.0 benchmark set is that it may be biased towards HiFi and Element as a result of assembly input and regions excluded. The HG002 T2T Q100 effort (https://github.com/marbl/HG002) to polish diploid assemblies will fill this gap and enable inclusion of additional homopolymers and tandem repeats in the benchmark that are noisy in HiFi. Because the current benchmark relies on GRCh38, it excludes regions with errors or missing sequence in GRCh38, and where HG002 has large structural changes relative to GRCh38. To benchmark the complex variation that occurs in regions not represented in GRCh38, approaches to represent and compare complex variation, such as

pangenomes[11,27], are needed. This manuscript provides an important initial benchmark from two T2T chromosomes, and highlights the strengths of this approach as well as remaining challenges for T2T benchmarks for the whole genome.

## Methods

This research complies with all relevant ethical regulations, and was approved by the NIST Research Protections Office (Protocol MML-009). This individual (male with X and Y sex chromosomes) was consented under the Harvard Personal Genome Project[12]. DNA extracted from a single large batch of cells is publicly available in National Institute of Standards and Technology Reference Materials 8391 (HG002) and 8392 (HG002 with parents), available at https://www.nist.gov/srm. DNA is extracted from publicly available cell lines GM24385 (HG002, RRID:CVCL_1C78) at the Coriell Institute for Medical Research National Institute for General Medical Sciences cell line repository.

### Benchmark set generation

Based on previous GIAB work[8], we included only loci with exactly one contig aligning from each haplotype (except in the X and Y non-PAR), which is the dip.bed file output from dipcall. For dipcall v0.3, we used the custom minimap2 (v2.24) parameter -z200000,10000 to align across larger SVs and more divergent regions like the MHC[28]. Furthermore, we excluded structural variants at least 50 bp in size and associated repeats, large repeats (segmental duplications, tandem repeats longer than 10 kbp, and satellites) that have any breaks in the assembly-assembly alignment, and regions around gaps in GRCh38[14]. This was implemented as a snakemake pipeline (https://github.com/nate-d-olson/defrabb). The version of the defrabb repo used to generate the XY benchmark was https://github.com/nate-d-olson/defrabb/tree/b0f08b6b0514555570e8f90fa51b0a86a3c904da. The config files used for the defrabb run, include input files for the repeats, are at https://github.com/nate-d-olson/defrabb/blob/b0f08b6b0514555570e8f90fa51b0a86a3c904da/config/analyses_20230315_v0.011-HG002XY.tsv and https://github.com/nate-d-olson/defrabb/blob/b0f08b6b0514555570e8f90fa51b0a86a3c904da/config/resources.yml.

We refined the benchmark using an active evaluation approach (https://github.com/usnistgov/active-evaluation). Based on curation of 12 callsets compared against the draft benchmark (see "Visualizing and curating variants to understand errors" of Olson et al[15].), we further excluded homopolymers longer than 30 bp, tandem repeats discordant with a different HG002 assembly, homopolymers discordant with Element avidity-based sequencing variant calls, as well as regions with ambiguous or incorrect assembled sequence or assembly-assembly alignments.

### Benchmark set evaluation

We evaluated the benchmark to test our criteria for Reference Materials to be fit-for-purpose, which in this case is the reliable identification of errors in diverse variant callsets[15]. In this work, we piloted a method for evaluating machine learning systems called Active Evaluation that takes advantage of stratifying a dataset and estimates a confidence interval for individual strata. We defined 15 stratifications (Fig. 3A) based on homopolymer length, difficult-to-map regions, and pseudoautosomal regions, as well as SNVs vs indels and putative false positives versus false negatives. We additionally performed long range PCR followed by Sanger sequencing on a subset of variants in challenging genes in chromosomes X and Y as a means of orthogonal validation of the benchmark variants (Supplementary Data 3).

### Statistics and reproducibility

Because this benchmark is for a single individual with extensive sequencing data, no statistical method was used to predetermine sample size. No data were excluded from the analyses. The experiments were not randomized. The Investigators were not blinded to allocation during experiments and outcome assessment.

### Reporting summary

Further information on research design is available in the Nature Portfolio Reporting Summary linked to this article.

## Data availability

The benchmark vcf and bed, as well as supporting files, are available at [https://ftp.ncbi.nlm.nih.gov/ReferenceSamples/giab/release/AshkenazimTrio/HG002_NA24385_son/chrXY_v1.0/]. The sequencing data used in this study are available in the NCBI SRA database under accession code PRJNA200694.

## Code availability

The code used to generate the XY benchmark is at [https://github.com/nate-d-olson/defrabb/tree/b0f08b6b0514555570e8f90fa51b0a86a3c904da]. The config files used for the defrabb run, include input files for the repeats, are at [https://github.com/nate-d-olson/defrabb/blob/b0f08b6b0514555570e8f90fa51b0a86a3c904da/config/analyses_20230315_v0.011-HG002XY.tsv] and [https://github.com/nate-d-olson/defrabb/blob/b0f08b6b0514555570e8f90fa51b0a86a3c904da/config/resources.yml]. We refined the benchmark using an active evaluation approach available at [https://github.com/usnistgov/active-evaluation]. The code to create the exclusion bed files is at [https://github.com/jmcdani/giab-chrXY-benchmark/blob/main/scripts/chrXY_benchmark_exclusions.ipynb].

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

## Acknowledgements

Certain commercial equipment, instruments, or materials are identified to specify adequately experimental conditions or reported results. Such identification does not imply recommendation or endorsement by the National Institute of Standards and Technology, nor does it imply that the equipment, instruments, or materials identified are necessarily the best available for the purpose. C.X. was supported by the National Center for Biotechnology Information of the National Library of Medicine (NLM), National Institutes of Health. J.W., J.M., N.D.O., C.G., P.F., and J.M.Z. were supported by intramural funding at the National Institute of Standards and Technology.

## Author contributions

J.W., M.F.E.M., U.S.E., F.J.S., L.M., K.S., A.C., M.B.-B., D.F., B.J.P., S.M.E.S., D.J., A.M.-B., L.A.R.-R., J.M.L.-S., C.F., G.N., J.A.L., P.C.B., M.W., J.M., and J.M.Z. curated and evaluated the benchmark. J.W., C.G., P.F., and J.M. performed the active evaluation. L.H. performed Sanger confirmation. J.W., N.D.O., J.M., and J.M.Z. analyzed the data to create the benchmark. C.X. and N.D.O. managed the data. All authors contributed to manuscript writing and editing.

## Competing interests

JAL is an employee of PacBio. SMES is an employee of Roche Sequencing Solutions. DF is an employee of Sentieon, Inc., and holds stock options as part of the standard compensation package. PCB sits on the Scientific Advisory Boards of Intersect Diagnostics Inc., Sage Bionetworks and BioSymetrics Inc. LM is an employee and shareholder of Illumina Inc. KS and AC are employees of Google LLC and own Alphabet stock as part of the standard compensation package. FJS has support from ONT, Illumina, Pacbio and Genentech. The remaining authors declare no competing interests.
