## [Peer Review file · Nature Communications]

Small variant benchmark from a complete assembly of X and Y chromosomes

Corresponding Author: Dr Justin Zook

Version 0:

Reviewer comments:

Reviewer #1

(Remarks to the Author)

The authors present a comprehensive approach to benchmarking small variants in the sex chromosomes, which are known for their complex repetitive regions and medical significance. They developed a benchmark set containing 111,725 variants for the Genome in a Bottle HG002 reference material using complete, polished de novo assemblies of the X and Y chromosomes. They introduced a new active evaluation method to ensure the benchmark accurately identifies errors in challenging genomic regions and across various sequencing technologies. The study highlights the inclusion of difficult regions such as long homopolymers, tandem repeats, and gene conversions, while addressing the remaining challenges in benchmarking complex structural variants. Validation through long-range PCR and Sanger sequencing confirmed the reliability of the benchmark. The research underscores the ongoing need for improved tools and standards for genome assembly and variant representation, marking a significant advancement in genomic benchmarking.

Strength: This manuscript presents a significant advancement in the benchmarking of small variants on the X and Y chromosomes. The methodologies are robust, the results are comprehensive, and the discussion provides valuable insights into future directions for genomic benchmarking.

A minor comment for further improvement:

Although the methodology is detailed, it might be helpful to include a flowchart or diagram summarizing the process of benchmark generation and evaluation. This would provide a quick visual reference for readers.

Reviewer Response

REVIEWERS' COMMENTS

Reviewer #1 (Remarks to the Author):

The authors present a comprehensive approach to benchmarking small variants in the sex chromosomes, which are known for their complex repetitive regions and medical significance. They developed a benchmark set containing 111,725 variants for the Genome in a Bottle HG002 reference material using complete, polished de novo assemblies of the X and Y chromosomes. They introduced a new active evaluation method to ensure the benchmark accurately identifies errors in challenging genomic regions and across various sequencing technologies. The study highlights the inclusion of difficult regions such as long homopolymers, tandem repeats, and gene conversions, while addressing the remaining challenges in benchmarking complex structural variants. Validation through long-range PCR and Sanger sequencing confirmed the reliability of the benchmark. The research underscores the ongoing need for improved tools and standards for genome assembly and variant representation, marking a significant advancement in genomic benchmarking.

Strength: This manuscript presents a significant advancement in the benchmarking of small variants on the X and Y chromosomes. The methodologies are robust, the results are comprehensive, and the discussion provides valuable insights into future directions for genomic benchmarking.

A minor comment for further improvement:

Although the methodology is detailed, it might be helpful to include a flowchart or diagram summarizing the process of benchmark generation and evaluation. This would provide a quick visual reference for readers.

Thank you for this helpful suggestions. We have added a new Figure 1 as suggested.